# Psychosocial Factors and Obesity in Adolescence: A Case-Control Study

**DOI:** 10.3390/children8040308

**Published:** 2021-04-18

**Authors:** Elisabeth K. Andrie, Marina Melissourgou, Alexandros Gryparis, Elpis Vlachopapadopoulou, Stephanos Michalacos, Anais Renouf, Theodoros N. Sergentanis, Flora Bacopoulou, Kyriaki Karavanaki, Maria Tsolia, Artemis Tsitsika

**Affiliations:** 1MSc Program “Strategies of Developmental and Adolescent Health”, Second Department of Pediatrics, P. & A. Kyriakou Children’s Hospital, School of Medicine, National and Kapodistrian University of Athens, 11527 Athens, Greece; anais.renouf@hotmail.com (A.R.); tsergentanis@yahoo.gr (T.N.S.); info@youth-health.gr (A.T.); 2Department of Endocrinology & Metabolism—Diabetology Center, Korgialenio Benakio—Hellenic Red Cross General Hospital of Athens, 11526 Athens, Greece; melmarina2004@yahoo.gr; 3Department of Diabetes Mellitus and Metabolism, School of Medicine, National and Kapodistrian University of Athens, Aretaieion Hospital, 11528 Athens, Greece; al.grip@gmail.com; 4Department of Endocrinology, Growth and Development, P. & A. Kyriakou Children’s Hospital, 11527 Athens, Greece; elpis.vl@gmail.com (E.V.); stmichalakos@gmail.com (S.M.); 5Department of Clinical Therapeutics, National and Kapodistrian University of Athens, Alexandra Hospital, School of Medicine, 11528 Athens, Greece; 6Center for Adolescent Medicine and UNESCO Chair on Adolescent Health Care, First Department of Pediatrics, School of Medicine, National and Kapodistrian University of Athens, Aghia Sophia Children’s Hospital, 11527 Athens, Greece; bacopouf@hotmail.com; 7Diabetes and Metabolism Clinic, Second Department of Pediatrics, School of Medicine, National and Kapodistrian University of Athens, P. & A. Kyriakou Children’s Hospital, 11527 Athens, Greece; kkarav@yahoo.gr; 8Second Department of Pediatrics, School of Medicine, National and Kapodistrian University of Athens, P. & A. Kyriakou Children’s Hospital, 1527 Athens, Greece; matsolia@med.uoa.gr

**Keywords:** adolescents, obesity, psychosocial factors, psychological stress, children’s environment

## Abstract

Introduction: The continuously increasing prevalence of childhood obesity is reaching epidemic proportions. Greece is among the countries with the highest childhood obesity prevalence rates. The present study aims to identify psychosocial factors associated with excess body weight of adolescents. Methods: This case-control study was conducted in Athens, Greece, and included 414 adolescents aged 11–18 years. Anthropometric measurements were recorded, and an anonymous self-completed questionnaire captured the psychosocial background, family environment, peer relations, and school environment. Results: Of the total sample of adolescents, 54.6% had normal body weight and 45.4% were overweight or obese. A multivariate logistic regression analysis showed that the factors related to the presence of overweight/obesity were adolescents’ age (OR = 0.416, *p* < 0.001), area of residence, presence of anxiety (OR = 4.661, *p* = 0.001), presence of melancholia (OR = 2.723, *p* = 0.016), participation in sports (OR = 0.088, *p* <0.001), smoking (OR = 0.185, *p* = 0.005), and mother’s occupation (OR = 0.065, *p* < 0.001). Conclusion: Psychological problems, maternal occupation, the absence of physical activity, and poor school performance were associated with adolescent overweight/obesity. It is important that screening for the presence of psychosocial issues is included in childhood obesity policies and treatment.

## 1. Introduction

Childhood and adolescent obesity is considered to be a major public health problem of the 21st century that has reached epidemic proportions [1]. During the last decade, its prevalence has increased, as the number of overweight and obese children has risen dramatically from 4% in 1975 to 18% in 2016 [2]. In 2016, approximately 340 million children and adolescents aged between five and 19 years old worldwide were diagnosed with overweight or obesity [2]. Boys tend to be overweight or obese more frequently than girls; among children and adolescents aged between five and 17 years, 22.9% of boys and 21.4% of girls were overweight or obese in the countries of the Organization for Economic Cooperation and Development (OECD) [3]. Greece, Italy, and Spain are among the countries with the highest childhood obesity rates in Europe [4]. Previous research documents the prevalence of overweight and obesity in Greek children, varying between 30–40% [5]. Other studies report rates of 37% for girls and 45% for boys for overweight or obesity in Greece [3].

The etiology of obesity is multifactorial. Genetic and environmental factors include certain infections, lifestyle, and eating behaviors. [2,6,7]. Psychosocial issues may also contribute to the development of obesity. During emotional or physical stress, the hypothalamic–pituitary–adrenal (HPA) axis is activated, while dopamine may also be involved [8]. Stress is associated with a change in eating behaviors; approximately 40% of people increase their food intake in time of stress. During stress periods, highly palatable foods, which are usually rich in sugar and fat, are consumed regardless of the presence of hunger [9].

Children may develop psychological stress due to physical, emotional, or sexual abuse as well as emotional or physical negligence [10]. Besides dysfunction within the family, lack of friends, bullying, and the perception of non-integration in the neighborhood can also lead to stress, depression, and low self-esteem [11]. Bullying is a type of aggression that can take place in any human relationship. Examples of adolescents who may be targeted are those who seem to be different from their peers because of their race, clothing, or weight status, but also because of their anxiety, low self-esteem, or disabilities. Furthermore, discrepancies in regard to social level and parental income may trigger bullying among peers [12]. Adolescents who are victims of bullying are at high risk for adverse mental health outcomes, such as low self-esteem [13], depression, anxiety, and suicide [14]. Psychological trauma during childhood is one of the most significant predictors for the development of obesity [15,16].

Although previous research documents the fact that childhood obesity is associated with psychosocial problems [17], to our knowledge there are no published data addressing the association between psychosocial factors and overweight/obesity in Greek adolescents. The aim of the present study was to identify differences in the psychosocial background between adolescents with normal weight and adolescents with overweight/obesity, as well as associations of psychosocial issues with excess body weight, among adolescents visiting a tertiary children’s hospital, in Athens, Greece.

## 2. Participants and Methods 

This case-control study drew data from adolescents aged 11–18 years who attended the tertiary “P. & A. Kyriakou” Children’s Hospital in Athens, Greece, in 2017 and 2018. The group of cases was recruited from the Adolescent Health Unit (A.H.U.) of the Second Department of Paediatrics of the “P. & A. Kyriakou” Children’s Hospital and consisted of overweight or obese adolescents who approached the Unit for that issue as new clients. The control group of adolescents with normal weight was recruited from outpatient services of the same hospital. They were attending for mild conditions such as mild respiratory conditions, gastrointestinal or genitourinary conditions, and various manifestations of allergy that are not related to obesity. Adolescents with severe underlying medical conditions, underweight or overweight adolescents, and those receiving chronic medication were excluded from the control group. The recruitment of normal-weight adolescents who attended the A.H.U. over the study period was avoided, as the majority of them had underlying psychological issues and chronic medical conditions that had affected their body weight.

Signed informed consent was obtained from the participating adolescents and their parents or legal guardians. The study was approved by the “P & A Kyriakou” Children’s Hospital Ethics Committee.

### 2.1. Data Collection Procedure

Findings from the clinical examination and anthropometric measurements of participants were recorded. Height was measured using a SECA 217 stadiometer, weight was measured with a Tanita Total Body Composition Analyzer TBF-410GS, and the body mass index (BMI) was calculated. Participants’ BMI classification was carried out in accordance with the International Obesity Task Force cut-off points for age and gender [18]. All anthropometric measurements were carried out by pediatricians. 

All participants completed the Achenbach Youth Self-Report questionnaire for children and teenagers (11 to 18 years) to assess their psychological profiles. The Youth Self-Report (YSR) is an instrument measuring psychosocial wellbeing as well as adolescents’ competence and problems in social, academic, cognitive, internalizing, and externalizing behaviors [17]. It has been standardized for use in Greece [19].

Participants’ school performance was classified as follows:Below the base: < 10 (for middle school and high school) or <5 (for primary school).Below average: 10–14.5 (for middle school and high school) or 5–6 (for primary school).Average: 14.5–16 (for middle school and high school) or 7–8 (for primary school).Above average: 16–20 (for middle school and high school) or 9–10 (for primary school). 

All measurements were recorded, and instruments were administered upon entry to the study by staff who had received appropriate training.

### 2.2. Statistical Analysis

Variables that are normally distributed are presented as mean ± SD. Categorical variables are presented as absolute and relative frequencies (%). In order to investigate whether two categorical variables were related, Pearson’s chi-square test was used. The Mann–Whitney U test was also used to compare the medians of two independent samples. To investigate possible confounding by sex, the same analysis was performed stratified by gender.

Multivariate logistic regression analysis was then performed to confirm which parameters were significantly associated with the presence of overweight or obesity. Results with a two-sided *p*-value < 0.05 were considered statistically significant, whereas results with a two-sided *p*-value between 0.05 and 0.10 were considered as suggestive. All statistical analyses were performed using IBM SPSS v.23 (IBM Corp. Released2015. IBM SPSS Statistics for Windows, Version 23.0, IBM Corp., Armonk, NY, USA) software.

## 3. Results

Initially, 573 adolescents were recruited, but 159 were subsequently excluded from the analysis due to incomplete data.

Thus, a total of 414 adolescents with mean age (±SD) of 15.09 ± 1.81 years participated in this study. Among them, 233 (56.3%) were girls and 181 (43.7%) were boys. The mean weight (± SD) of the participants was 68.63 ± 16.57 kg, the mean height (±SD) was 1.67 ± 0.08 m, and the mean BMI (± SD) was 24.54 ± 5.56 kg/m^2^. In terms of their BMI, 54.6% had a normal BMI, 20.3% were in the overweight range, and 25.1% were in the obese range. 

The demographic data of the participants, according to BMI categories, are shown in Table 1. In the overweight–obese group, boys made up only one third of the overweight but about half of the obese (*p* = 0.008). The average age (± SD) differed significantly (*p*-value < 0.001) between participants with normal weight (15.8 ± 1.3) and adolescents with overweight or obesity (14.3 ± 2.0). Regarding maternal occupation, household status was reported for the majority (30.7%) of normal weight participants and “public sector employee” for the majority (28.6%) of participants with overweight or obesity (*p* = 0.009). Most of the participants (96.1%) lived in the Attica Region, and only 3.9% lived in other areas of Greece; 38.9% of normal weight participants were living in the western suburbs, while 35% of the overweight/obese adolescents lived in the center of Athens (*p*-value < 0.001). The vast majority of adolescents were Greek (92.8%), while 29 (7.2%) had other nationalities, mostly Albanian. Regarding parental marital status, 82.4% of the participants with normal weight had married parents, while adolescents with overweight or obesity exhibited lower rates (*p* = 0.007).

Table 2 presents the psychosocial factors, in relation to BMI categories. Among study participants, 245 (66.6%) suffered from anxiety, which was more pronounced in the overweight/obese group (*p* < 0.001). Furthermore, 118 (32.6%) of the participants had melancholic depression (with overweight/obese adolescents exhibiting higher rates, *p* = 0.003), and 35 (9.6%) had suicidal behaviors. Additionally, 66 (18%) reported low self-esteem, with normal weight adolescents exhibiting higher rates (*p* < 0.001). Concerning bullying, 91 (25.3%) adolescents reported that they had been victims of bullying at least once in their life. 

The peer relations of the participants were also examined (Table 3). Among study participants, 170 (45.7%) had already been in a romantic relationship and 89 (24.9%) had complete sexual activity; significantly more adolescents with normal weight than those with overweight/obesity (*p* < 0.001). Additionally, 294 (77.6%) participants were participating in at least one sport activity, significantly more adolescents with normal weight (89.9%) than those with overweight/obesity (61.1%, *p*-value < 0.001). Concerning school performance (Table 4), most adolescents (281, 99.6%) were going to school, and only one (0.4%) did not attend school. There was a statistically significant difference in school performance between normal and overweight/obese adolescents (*p* < 0.001); although most of the participants were above average, more normal-weight participants had average grade (37.2%), while most participants with overweight or obesity (27.6%) were below average. Moreover, normal-weight participants reported more unjustified absences (49%) compared to adolescents with overweight or obesity (24.1%, *p* < 0.001). 

When stratified by sex, similar results were found between normal weight and adolescents with overweight or obesity for both girls and boys. Nevertheless, in terms of parental marital status, a statistically significant difference was observed between normal weight and boys with overweight or obesity (*p* = 0.006), which was not found in girls (*p*-value = 0.103). Additionally, maternal occupation differed significantly between the two BMI groups in girls (*p* = 0.045), with the majority (39.0%) of normal weight participants’ mothers being housewives and the majority of overweight/obese participants’ mothers being employees in the public or private sectors (30.2%). No significant differences in maternal occupation were found in boys according to their BMI (*p* = 0.187). Finally, in terms of school performance, statistically significantly more unjustified absences were reported by normal-weight girls vs. girls with overweight or obesity (42.5% vs 12.5% respectively, *p* = 0.002). No significant differences in unjustified absences in boys according to their BMI were observed.

A multivariate logistic regression analysis (Table 5) showed that the factors statistically related to the presence of overweight/obesity were younger age (OR = 0.416, *p* < 0.001), area of residence, presence of anxiety (OR = 4.661, *p* = 0.001) or melancholic depression (OR = 2.723, *p* = 0.016), sport’s activities (OR = 0.088, *p* < 0.001), smoking (OR = 0.185, *p* = 0.005), and maternal occupation (OR = 0.065, *p* < 0.001). Other parameters that were included in the model but were not related significantly to the presence of obesity were parental occupation, ethnicity, bullying, number of siblings, and romantic or sexual relationships.

## 4. Discussion

The prevalence of childhood overweight and obesity is increasing rapidly worldwide and is recognized as a leading threat to public health. The present study examined the psychosocial correlates of obesity in adolescents in Greece. Statistically significant differences between overweight/obese cases and controls were observed in terms of sex, maternal employment, parental marital status, anxiety, melancholic depression, low self-esteem, romantic and sexual relationships, sports, school performance, and school absenteeism.

In the present study, maternal employment was significantly associated with adolescents’ obesity. Thus, in the normal weight group, most mothers were unemployed, while in the overweight/obese group most of them were public sector employees. On the other hand, no relation between overweight/obesity and paternal employment was observed. Our results concerning mothers’ employment status are consistent with several studies [20,21,22,23,24,25] that have linked maternal employment to children’s and adolescents’ obesity. In addition, Anderson et al. observed that the more hours the mothers worked, the higher the risk for the children to become overweight or obese [25]. Nevertheless, the mechanisms that mediate these associations remain largely unknown. The main channels associated with greater weight include less time allocated to housework (including meal preparation) and a reduction in maternal supervision regarding children’s food intake and physical activity [26,27,28]. The present study showed that a significantly higher percentage (28%) of overweight and obese adolescents than normal-weight participants (15.3%) had divorced parents. The GENDAI study carried out in Greece confirmed that parental marital status plays a key role in the emergence of obesity in adolescents [28]. This was confirmed not only in Greece, where traditional family status is more frequent, but also in studies from other European countries, such as Poland, the United Kingdom, Iceland, and Sweden [20,29,30,31,32], indicating that a stable family environment is important for the preservation of normal body weight [33]. Research has indicated that children of single parents are less likely to eat at the table together with the parent and are allowed to play and watch television during meals [34]. Children of single-parent households are reported to consume more total fat, saturated fat, and sweetened beverages and also watch television/video for more than two hours daily more frequently when compared to children of two-parent family households [35]. On the other hand, a similar study from Nordic countries did not confirm our results [31].

In the present study, anxiety was also linked to higher BMI in both genders. This result confirmed the findings of previous studies [25,36,37] that revealed a gender difference in the link between anxiety and the development of obesity. In particular, most of them identified a stronger link between the development of overweight and obesity due to anxiety in girls [25,36]. Separation anxiety was associated with increased waist circumference and BMI in boys, whereas in girls, somatic symptoms of anxiety were associated with waist circumference and higher body fat [25].

Previous research has demonstrated associations between obesity and depression in children and adolescents [38]. Nevertheless, the mental well-being and psychiatric health of children and adolescents suffering from obesity are the subject of considerable debate [25,36,39,40,41,42,43,44,45]. There are two systematic reviews and a meta-analysis on this topic [37,45,46] suggesting that obese children and adolescents are more likely to suffer from depression and depressive symptoms, with females being at higher risk. Consistent with previous reports, our study indicated that melancholic depression was related to overweight/obesity in adolescents [25,40,41,47,48]. There are three possible pathways that could account for these disorders. Obesity could lead to depression through weight stigma [49], poor self-esteem [50], and/or reduced mobility and ability to engage in activities [51]. Depression could lead to obesity directly through the occurrence of depressive symptoms (e.g., increased appetite, poor sleep, lethargy resulting in decreased calorie expenditure, and/or reduced energy to obtain and cook healthy foods), antidepressant medication side effects, or attempts to self-medicate depressive feelings with unhealthy foods [51,52,53]. Further investigation of the mechanisms underlying the observed comorbidity is needed.

We found that the frequency of participation in sport activities was significantly higher in normal-weight than overweight/ obese adolescents. This is expected and has also been demonstrated in previous studies, as a lack of physical activity in adolescence has been found to lead to obesity [54,55]. There is a bidirectional effect between the lack of physical activity and increased BMI, as the lack of physical activity may lead to an increase in BMI, but inversely, an increased BMI may lead to reduced participation in sport activities [55]. In addition, adolescents may also be more self-conscious about their physical appearance and thus refrain from exercising in front of others [56]. In this survey, apart from assessing the frequency of physical activity among adolescents, we also found that those who were socially integrated and participated in team sports had a lower probability of being overweight because of higher self-esteem and better relationships with their peers.

In our study, both school performance and unjustified absences were associated with overweight/obesity. To our knowledge, there is no similar research identifying a link between school truancy and development of overweight/obesity. In addition, poor school performance may be related to obesity. Poor school performance is associated with negative feelings of failure and inability to succeed. These in turn are related to depressive symptoms, worrying about school results, and overeating [57].

To our knowledge, this is the first study to examine whether psychosocial factors are associated with increased prevalence of overweight and obesity among Greek adolescents. One limitation of the study is the relatively small sample, which in addition is not representative of the population of all of Greece, as it is restricted to adolescents living in the Attica Region, although this does represent 35% of the country’s population.

In conclusion, this study showed that psychological problems, such as anxiety and melancholic depression, are associated with obesity. Moreover, maternal occupation, the absence of physical activity, and poor school performance were associated with adolescent overweight/obesity. Therefore, it is of great importance that screening for the presence of psychosocial issues should be included in childhood obesity policies and proper handling of these issues should be provided. In addition, public health policies should be formulated and strengthened in the future targeting physical activity, maternal employment, and work schedules early in adolescence, with special attention to girls.

## Figures and Tables

**Table 1 children-08-00308-t001:** Participants’ demographic data by weight status (normal weight, overweight/obese, overweight and obese separately).

Variables	Normal Weight (*N* = 226) Frequency (%)	Overweight-Obese (*N* = 188) Frequency (%)	*p*-Value *	Overweight (*N* = 84) Frequency (%)	Obese (*N* = 104) Frequency (%)	*p*-Value *
Sex						
Female	126 (55.8%)	107 (56.9%)	0.812	57 (67.9%)	50 (48.1%)	0.008
Male	100 (44.2%)	81 (43.1%)		27 (32.1%)	54 (51.9%)	
Age						
mean (±SD)	15.80 (±1.26)	14.26 (±2.00)	<0.001	14.59 (±2.04)	13.98 (±1.94)	0.047
Siblings						
0	30 (13.3%)	34 (20.5%)	0.033	15 (21.4%)	19 (19.8%)	0.518
1	144 (63.7%)	97 (58.4%)		39 (55.7%)	58 (60.4%)	
2	32 (14.2%)	29 (17.5%)		11 (15.7%)	18 (18.8%)	
3	13 (5.8%)	6 (3.6%)		5 (7.1%)	1 (1.0%)	
4	4 (1.8%)	0 (0.0%)		0 (0.0%)	0 (0.0%)	
5	3 (1.3%)	0 (0.0%)		0 (0.0%)	0 (0.0%)	
Paternal occupation						
Unemployed	5 (2.3%)	7 (3.9%)	0.866	3 (3.7%)	4 (4.1%)	0.836
Household	1 (0.5%)	0 (0.0%)		0 (0.0%)	0 (0.0%)	
Private sector employee	79 (36.6%)	71 (39.4%)		32 (39%)	39 (39.8%)	
Public sector employee	42 (19.4%)	34 (18.9%)		15 (18.3%)	19 (19.4%)	
Self-employed	74 (34.3%)	58 (32.2%)		29 (35.4%)	29 (29.6%)	
Retired	15 (69%)	10 (5.6%)		3 (3.7%)	7 (7.1%)	
Maternal Occupation						
Unemployed	6 (2.8%)	13 (7%)	0.009	4 (4.8%)	9 (8.8%)	0.552
Household	66 (30.7%)	32 (17.3%)		13 (15.7%)	19 (18.6%)	
Private sector employee	61 (28.4%)	52 (28.1%)		21 (25.3%)	31 (30.4%)	
Public sector employee	53 (24.7%)	53 (28.6%)		27 (32.5%)	26 (25.5%)	
Self-employed	27 (12.6%)	29 (15.7%)		16 (19.3%)	13 (12.7%)	
Retired	2 (0.9%)	6 (3.2%)		2 (2.4%)	4 (3.9%)	
Parental Marital Status						
Married	183 (82.4%)	111 (68.9%)	0.007	48 (70.6%)	63 (67.7%)	0.677
Divorced	34 (15.3%)	45 (28%)		19 (27.9%)	26 (28%)	
Death of a parent	5 (2.3%)	5 (3.1%)		1 (1.5%)	4 (4.3%)	
Recidency Area						
Athens Center	0 (0.0%)	65 (35.1%)	<0.001	22 (26.8%)	43 (41.7%)	0.506
North Suburbs	0 (0.0%)	29 (15.7%)		14 (17.1%)	15 (14.6%)	
South Suburbs	37 (16.4%)	22 (11.9%)		10 (12.2%)	12 (11.7%)	
East Suburbs	25 (11.1%)	21 (11.4%)		13 (15.9%)	8 (7.8%)	
West Suburbs	88 (38.9%)	15 (8.1%)		7 (8.5%)	8 (7.8%)	
Piraeus	64 (28.3%)	15 (8.1%)		7 (8.5%)	8 (7.8%)	
Rest of Attica	11 (4.9%)	3 (1.6%)		1 (1.2%)	2 (1.9%)	
Rest of Greece	1 (0.4%)	15 (8.1%)		8 (9.8%)	7 (6.8%)	
Nationality						
Greek	200 (90.1%)	176 (96.2%)	0.191	79 (97.5%)	97 (95.1%)	0.829
Albanian	15 (6.8%)	6 (3.3%)		2 (2.5%)	4 (3.9%)	
Russian	2 (0.9%)	0 (0.0%)		0 (0.0%)	0 (0.0%)	
African countries	1 (0.5%)	0 (0.0%)		0 (0.0%)	0 (0.0%)	
Other European countries	3 (1.4%)	1 (0.5%)		0 (0.0%)	1 (1.0%)	
Asian countries	1(0.5%)	0 (0.0%)		0 (0.0%)	0 (0.0%)	

* Statistical analysis was performed with the Chi-Square test.

**Table 2 children-08-00308-t002:** Participants’ psychosocial factors by weight status (normal weight, overweight/obese, overweight and obese separately).

Variables	Normal Weight (*N* = 226) Frequency (%)	Overweight-Obese (*N* = 188) Frequency (%)	*p*-Value *	Overweight (*N* = 84) Frequency (%)	Obese (*N* = 104) Frequency (%)	*p*-Value *
Anxiety						
Yes	116 (55.5%)	129 (81.1%)	<0.001	56 (83.6%)	73 (79.3%)	0.544
No	93 (44.5%)	30 (18.9%)		11 (16.4%)	19 (20.7%)	
Melancholic Depression						
Yes	55 (26.3%)	63 (41.2%)	0.003	26 (42.6%)	37 (40.2%)	0.867
No	154 (73.7%)	90 (47.9%)		35 (57.4%)	55 (59.8%)	
Suicidal behavior						
Yes	23 (11%)	12 (7.8%)	0.371	5 (8.1%)	7 (7.7%)	1.000
No	187 (89%)	141 (92.2%)		57 (91.9%)	84 (92.3%)	
Low self-esteem						
Yes	57 (27%)	9 (5.8%)	<0.001	3 (4.7%)	6 (6.5%)	0.738
No	154 (73%)	147 (94.2%)		61 (95.3%)	86 (93.5%)	
Bullying						
Yes	57 (27%)	34 (22.8%)	0.391	9 (15%)	25 (28.1%)	0.074
No	154 (73%)	115 (77.2%)		51 (85%)	64 (71.9%)	

* Statistical analysis was performed with the Chi-Square test.

**Table 3 children-08-00308-t003:** Participants’ activities by weight status (normal weight, overweight/obese, overweight and obese separately).

Variables	Normal Weight (*N* = 226) Frequency (%)	Overweight-Obese (*N* = 188) Frequency (%)	*p*-Value *	Overweight (*N* = 84) Frequency (%)	Obese (*N* = 104) Frequency (%)	*p*-Value *
Romantic relationships						
Yes	97 (51.6%)	73 (47.1%)	<0.001	35 (53.8%)	38 (42.2%)	0.192
No	91 (48.4%)	82 (52.9%)		30 (46.2)	52 (57.8%)	
Sexual relationships						
Yes	75 (36.2%)	14 (9.3%)	<0.001	7 (11.3%)	7 (8.0%)	0.573
No	132 (63.8%)	136 (90.7%)		55 (88.7%)	81 (92%)	
Sports activities						
Yes	195 (89.9%)	99 (61.1%)	<0.001	48 (69.6%)	51 (54.8%)	0.073
No	22 (10.1%)	63 (38.9%)		21 (30.4%)	42 (45.2%)	
Hobbies						
Yes	195 (89.9%)	141 (92.2%)	0.473	59 (90.8%)	82 (93.2%)	0.762
No	22 (10.1%)	12 (7.8%)		6 (9.2%)	6 (6.8%)	

* Statistical analysis was performed with the Chi-Square test.

**Table 4 children-08-00308-t004:** Participants’ school performance by weight status (normal weight, overweight/obese, overweight and obese separately).

Variables	Normal Weight (*N* = 226) Frequency (%)	Overweight-Obese (*N* = 188) Frequency (%)	*p*-Value *	Overweight (*N* = 84) Frequency (%)	Obese (*N* = 104) Frequency (%)	*p*-Value *
School performance						
Below the base	2 (1.0%)	0 (0.0%)	<0.001	0 (0.0%)	0 (0.0%)	0.026
Below average	12 (5.8%)	37(27.6%)		9 (15.5%)	28 (36.8%)	
Average	77 (37.2%)	25(18.7%)		13 (22.4%)	12 (15.8%)	
Above average	116 (56%)	72 (53.7%)		36 (62.1%)	36 (47.4%)	
Unjustified absences						
Yes	102 (49.0%)	14 (24.1%)	0.001	6 (27.3%)	8 (22.2%)	0.756
No	106 (51.0%)	44 (75.9%)		16 (72.7%)	28 (77.8%)	

* Statistical analysis was performed with the Chi-Square test.

**Table 5 children-08-00308-t005:** Multivariate logistic regression results.

Variables	OR	*p*-Value	95% C.I. for OR	
			Lower	Upper
Participants’ age ^a^	0.416	<0.001	0.312	0.555
Sports activities ^a^ Yes	0.088	<0.001	0.035	0.220
Residency area ^a^				
West suburbs	0.062	<0.001	0.021	0.181
Piraeus	0.089	<0.001	0.032	0.248
Rest of Attica/Greece	1.084	0.899	0.314	3.737
Anxiety ^a^	4.661	0.001	1.837	11.829
Maternal occupation ^a^				
Housewife	0.065	0.001	0.012	0.344
Working	0.261	0.261	0.060	1.128
Retired	0.479	0.620	0.026	
Smoking ^a^ Yes	0.185	0.005	0.058	0.594
Melancholic Depression^a^	2.723	0.016	1.210	6.132
Hobbies? ^a^ Yes	0.488	0.259	0.140	1.698
Sex ^a^				
Boys	1.571	0.261	0.714	3.454
Parental marital status ^a^				
Married	0.764	0.536	0.326	1.790
Siblings ^a^	0.366	0.647	0.251	1.665

^a^ Reference levels; Sex: Girls; Parental marital status: Divorced/Death of a parent; Residency area: Center of Athens/North suburbs/South suburbs/East suburbs; Siblings: No; Sports: No; Smoking: No; Hobbies: No; Maternal occupation: Unemployed.

## Data Availability

The data presented in this study are available on request from the corresponding author.

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
