# Peer review of "Psychosocial Factors and Obesity in Adolescence: A Case-Control Study"

_children, 2021, doi:10.3390/children8040308_

Round 1

Reviewer 1 Report

Introduction

  • The authors suggest that there is little known about how psychosocial factors affect childhood obesity, however, there have been many studies that have looked at this link. Also, the current study doesn’t demonstrate causation
  • Page 2, line 51 – the term “rocketed” is too theatrical-sounding. I would consider something like “dramatically increased”?
  • Page 2, line 56 – what is OECD? This should be spelled out
  •  

Methods

  • This section feels underdeveloped. More information is needed about the sample and the measures
  • The International Obesity Task Force (IOTC) should be spelled out rather than abbreviated
  • Page 2, lines 95 – 96 – I think a period is needed after Children’s Hospital.
  • Page 3, line 108 – the authors say they used the Tanita Total Body Composition Analyzer which assesses body fat percentage. Why isn’t this reported in the study? BMI doesn’t take into account fat vs lean tissue
  • I’m confused about the recruitment section. The authors say that overweight and obese adolescents from the AHU, and they mention that the normal weight adolescents were purposely not recruited from AHU, but were from other departments of the hospital? Wouldn’t they also be dealing with some type of medical condition anyway? Also, if the authors avoided this unit for normal weight adolescents due to the presence of psychological factors, wouldn’t it then be expected that the overweight/obese participants who did come from this unit would have these psychological issues, thus skewing the results? I don’t understand this particular approach. Please provide further detail.
  • Page 3, line 111 – where the procedures standardized among pediatricians? Were they all taken at the beginning of the study
  • The authors need to include more details regarding the Achenbach Youth Self-Report survey and cite the original validation study
  • Why did only the overweight group complete the HEEADSSS? That appears to be a major limitation of the study since you cannot compare them with the normal weight adolescents. Also, this should be spelled out before the abbreviation is used.
  • I’m confused with the way education was classified. Is this a validated classification method?
  • Page 3, line 128 – 129 – the authors mention qualitative data but none of the methods or results suggest anything qualitative was collected

Results

  • Page 4, line 149 – The authors say that “most were female”, but that does not appear to be the case as it looks like the sample is pretty split. Additionally, do the authors mean to say that “a greater % of females were classified as overweight compared to males”?
  • Page 7, lines 219 – 222. The font appears to differ from the rest of the manuscript
  • I’m confused by page 8, lines 32 – 35. Not sure why this is formatted like this?

Discussion

  • Page 10, lines 310 – 311 – this isn’t actually accurate as many studies have examined psychosocial parameters and increased prevalence of overweight and obesity among adolescents. Could the authors mean the first study in Greece?
  • The authors fail to note several additional limitations to their study. The normal weight
  • How much do these results actually tell us since the authors claim that this isn’t representative of the Greek adolescent population, however, that was their reasoning for why this is a novel study (Last paragraph of the introduction)?

Reviewer 2 Report

You did a really good job. I'll give you some comments.

I am not sure if this is a case-control study. Since all survey points are the same, isn’t it a section analysis study? Please check the study design.

line 35: Results: The study sample consisted of 226 adolescents,

The normal weight group was 226, and the total sample was 414.

line 81: There is evidence that childhood obesity can lead to psychosocial problems,

Please add a reference

line 96: Hospital Adolescents with normal weight

Please put a period.... Hospital. Adolescents with~~

lines 102-103: A signed informed consent was obtained from the adolescents’ parents or legal 102 guardians...

Did you not get consent from the study participant (adolescents)?

lines 113-119: In addition, adolescents ~~~ psychosocial environment.

  Did you get HEEADSSS from only overweight or obesity adolescents? I haven’t seen anything about HEEDSSS in the text. If you have used this measurement, you should include it in the text. Or delete it.

There is a lack of description of tools for measuring depression or anxiety. Please add details such as what kind of person has developed it, how many questions, and what kind of reliability is it?

Line 145: 24.54 ± 5.56 kg/m2 [19]

Do you need a reference to describe the results?

Lines 153-154: “private employee”

28.6% are public employees.

Line 245: “private employee”

28.6% are public employees(table 1).

Lines 310-311: this is the first study to examine whether psychosocial parameters are associated with increased prevalence of overweight and obesity among adolescents.

It was first implemented in Greece, so you need to change to Greek youth.

Line 96: Hospital. Adolescents~~

Line 315: with obesity. Moreover~~

Pleases add a period.

Round 2

Reviewer 1 Report

The revised version is significantly improved, and the authors have addressed my concerns!

Author Response

We thank your positive comments